# RNA editing derived epitopes function as cancer antigens to elicit immune responses

Minying Zhang[1], Jens Fritsche [2], Jason Roszik [1], Leila J. Williams [1], Xinxin Peng[3], Yulun Chiu[3], Chih-Chiang Tsou[7], Franziska Hoffgaard[2], Valentina Goldfinger[2], Oliver Schoor[2], Amjad Talukder[1], Marie A. Forget[1], Cara Haymaker[1], Chantale Bernatchez[1], Leng Han [4], Yiu-Huen Tsang[5], Kathleen Kong[5], Xiaoyan Xu[3,6], Kenneth L. Scott[5], Harpreet Singh-Jasuja[2,7], Greg Lizee[1], Han Liang [3,8], Toni Weinschenk [2,7], Gordon B. Mills[8] & Patrick Hwu[1]

In addition to genomic mutations, RNA editing is another major mechanism creating sequence variations in proteins by introducing nucleotide changes in mRNA sequences. Deregulated RNA editing contributes to different types of human diseases, including cancers. Here we report that peptides generated as a consequence of RNA editing are indeed naturally presented by human leukocyte antigen (HLA) molecules. We provide evidence that effector CD8$^+$ T cells specific for edited peptides derived from cyclin I are present in human tumours and attack tumour cells that are presenting these epitopes. We show that subpopulations of cancer patients have increased peptide levels and that levels of edited RNA correlate with peptide copy numbers. These findings demonstrate that RNA editing extends the classes of HLA presented self-antigens and that these antigens can be recognised by the immune system.

[1] Department of Melanoma Medical Oncology, The University of Texas MD Anderson Cancer Center, 7455 Fannin Street, Unit 904, Houston, TX 77054, USA. [2] Immatics Biotechnologies, Paul-Ehrlich-Str 15, 72076 Tübingen, Germany. [3] Department of Bioinformatics and Computation Biology, The University of Texas MD Anderson Cancer Center, P. O. Box 301402 , Houston, TX 77230, USA. [4] Department of Biochemistry and Molecular Biology, The University of Texas Health Science Center at Houston McGovern Medical School, 6431 Fannin Street MSB 6.200, Houston, TX 77030, USA. [5] Department of Molecular and Human Genetics, Baylor College of Medicine, One Baylor Plaza, BCM225, Houston, TX 77030, USA. [6] Department of Pathophysiology, College of Basic Medicine, China Medical University, Shenyang, 77 Puhe Rd, Shenbei Xinqu, Shenyang Shi, 110122, China. [7] Immatics US, 2130W Holcombe Blvd, Houston, TX 77030, USA. [8] Department of Systems Biology, The University of Texas MD Anderson Cancer Center, 1515 Holcombe Blvd, Houston, TX 77230, USA. These authors contributed equally: Minying Zhang, Jens Fritsche. Correspondence and requests for materials should be addressed to T.W. (email: weinschenk@immatics.com) or to G.B.M. (email: gmills@mdanderson.org) or to P.H. (email: phwu@mdanderson.org)

T cells play a primary role in adaptive immunity against infections and tumorigenesis, and therapies based on manipulating T-cell activation have shown promise in cancer treatment[1–3]. These strategies include adoptive T-cell transfer (ACT) with tumour infiltrating lymphocytes (TIL) and checkpoint blockade with anti-CTLA4/anti-PD1 antibodies[3–5]. Cancer immunotherapy based on tumour-specific antigens that are recognised by tumour-reactive T cells has attracted more and more attention[6,7]. Tumour antigens include neo-antigens derived from patient-specific tumour mutations and self-antigens that are overexpressed in cancer[8]. While neo-antigens are derived from genomic mutations in tumour cells[7,9,10], protein variations might also result from RNA editing, a posttranscriptional process involving enzymatic modifications of specific nucleotides in RNA sequences[11,12]. The most common type of RNA editing, which converts adenosine to inosine (A→I editing), is catalysed by Adenosine Deaminases Acting on RNA (ADARs)[13,14]. The ADAR family includes 3 members: *ADAR1*, *ADAR2* and *ADAR3*. *ADAR1* and *ADAR2* play a major role in RNA editing with different target preference, while *ADAR3* negatively regulates the function of other ADARs[13,15,16]. Deregulated RNA editing contributes to different types of human diseases, including cancers[17–22]. Therefore, peptides derived from edited RNA transcripts—*edited peptides*—may be presented on human leukocyte antigen (HLA) and serve as a source for cancer antigens. However, despite this intriguing possibility, whether such peptides are indeed generated and capable of stimulating immune responses has so far been unknown.

The use of edited peptides in immunotherapy would require characterisation in three dimensions: peptide presentation, T-cell recognition and tumour association. First, the peptide should be processed and presented by the HLA class I antigen pathway in order to be considered as HLA ligand. Second, the peptide-HLA complex should be recognised by specific T-cells and be able to stimulate immune responses. Third, the peptide should show tumour-specific or tumour-associated presentation[23].

High-resolution mass-spectrometry has enabled identification and quantitation of HLA ligands that are naturally processed and presented. This elucidation of the immunopeptidomes involves immunoprecipitation followed by liquid chromatography–mass spectrometry (LC–MS) analysis of the eluted ligands[24]. Standard data analysis of LC-MS experiments relies on the comparison of MS spectra against theoretical spectra derived from a reference proteome. Due to the absence of editing information in reference proteomes, identification of edited peptides is usually missed and requires a specialised proteogenomics screening approach[25].

Here we report the identification and confirmation of five edited peptides from three editing sites by mass spectrometry. We report the in-depth characterisation of an editing site of cyclin I (*CCNI R75G*) with regard to peptide presentation, T-cell recognition and tumour association. We present relative quantitative data on primary healthy and tumour tissues supporting over-editing on peptide level for a subset of tumour patients. We confirm that peptide levels correspond to editing on RNA level by absolute quantitation. Further, we provide evidence that editded peptide specific T cells infiltrate into melanoma tumours and mediate killing of the tumour cells that express the edited antigens.

## Results
### Proteogenomics screening identifies edited HLA ligands.
To identify edited peptides, we designed a proteogenomics screening approach (Fig. 1a) that investigated data acquired by the antigen discovery platform XPRESIDENT®[24,26] that combines liquid chromatography-mass spectrometry (LC–MS) for identification and quantitation of HLA ligands with RNA sequencing (RNA-

seq) of corresponding mRNA from the same sample. HLA class I-peptides were isolated from 1514 different healthy and tumour samples from primary tissues of 1119 donors resulting in ~60 million fragment spectra (MS/MS). To search for edited peptides, we first constructed an RNA editome peptide database (Supplementary Data 1) derived from 1369 editing sites extracted from the Rigorously Annotated Database of A-to-I RNA editing (RADAR)[27]. Matching MS/MS spectra against the database identified 7 peptides (Supplementary Table 1) of which two were false positive. The other five sequences were experimentally confirmed to the largest extent possible which is necessary for novel sequences[28]. We conclusively confirmed the true positives by co-elution of corresponding synthetic isotope-labelled peptides using LC-MS (Supplementary Fig. 1).

Table 1 lists the five confirmed edited peptides of which four were derived from the well described editing sites *CCNI R75G* and *COPA I164V* and one from *CDK13 Q35R*, as previously identified by RNA-seq[27,29]. All peptides were extracted from primary tissue based on HLA-specific antibodies by immunoprecipitation (IP), thus ensuring binding and presentation of the peptides by HLA. The HLA restriction of each peptide was determined by HLA typing of each tissue (Table 1). Remarkably, each of the two peptides found for *CCNI* and *COPA* formed nested sets. For each edited peptide, the corresponding non-edited wild type was also detected.

### Increased levels of edited peptides in cancer subpopulations.
To investigate the tumour association of edited peptides, assessment of peptide presentation levels on a comprehensive data set of quantitative HLA peptidomics data is required. Thus, we focused on the HLA-A*02 ligands found for *CCNI* for in-depth characterisation making use of quantitative HLA-A*02 peptidome data for 925 samples covering tumour ($n = 504$) and normal tissues ($n = 421$). Both CCNI-WT peptides were detected in almost all A*02 positive samples, showing similar levels in normal and cancer tissues.While CCNI-ED was also presented on healthy tissues, it showed elevated abundances in several tumours (Fig. 1b). To identify samples with unusually high editing, the healthy sample population was used to define a normal reference range and the upper limit of normal (*ULN*) in particular. Samples with HLA peptide abundance above the *ULN* were defined as *over-edited*. Of the 504 tumours quantified, 36 (7%) showed over-editing of up to 40-fold above *ULN*. Considering individual tumour lineages, the most prevalent indications were ovarian cancer (41%, $n = 9/22$), melanoma (27%, $n = 4/15$) and breast cancer (22%, $n = 4/18$).

### Edited peptide-specific T cells infiltrate melanoma tumours.
Based on the described findings, we examined whether the detected HLA ligands are recognised by tumour-infiltrating lymphocytes (TILs) and if these TILs can kill cells presenting CCNI-ED. For this analysis, we took advantage of our unique resource of matched autologous pairs of melanoma TILs and tumour cell lines[4]. We synthesised edited *CCNI* peptides and their wildtype counterparts and evaluated their ability to activate TILs generated from human melanoma tumours. Remarkably, 3 out of 15 of the assessed HLA-A*02:01-expressing TILs (TIL2678, TIL2661 and TIL2559) strongly responded to CCNI-ED10, as demonstrated by robust production of the T-cell effector cytokine IFNγ in ELISPOT assays run in triplicates, whereas none of the TILs responded to CCNI-WT10 despite presentation on HLA (Fig. 2a). This finding suggested the existence of pre-existing effector T cells in tumour-infiltrating immune cells specifically reacting to CCNI-ED10 implying its in vivo function as antigenic epitope. A parallel assay using the same TILs identified one that reacted to CCNI-ED9 albeit more weakly than the T-cell

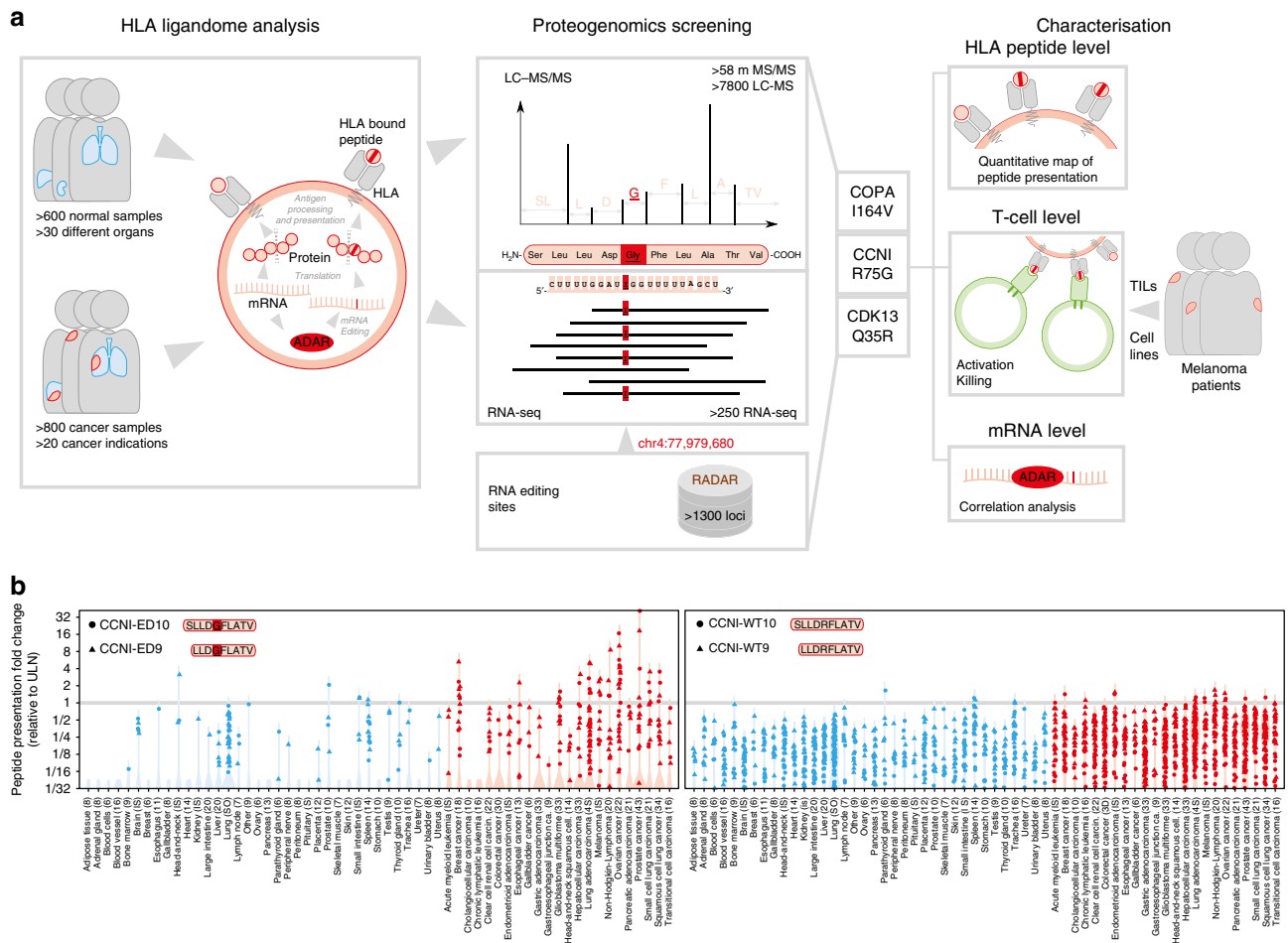

**Fig. 1** Discovery of RNA editing derived HLA peptides and MS-based quantitation of edited peptides from CCNI. **a** Pipeline combining RNA-seq and LC–MS data from primary tissue for discovery of HLA ligands derived from RNA editing sites listed in RADAR. Edited nucleotides or amino acids are underlined and highlighted in red. CCNI peptides were quantitatively analysed and compiled into an in vivo map of peptide abundance to assess tumour association. In parallel, deeper target characterisation by assessment of immunogenicity and T cell killing was performed. For further validation, correlation between peptide and mRNA levels of edited CCNI and ADAR were assessed. **b** Relative abundance of HLA-bound peptides derived from edited and non-edited wild type (WT) CCNI peptides isolated from tumour (red) and normal samples (blue). Each dot represents a sample for which the peptide was detected. Samples are grouped according to healthy organ or tumour indication. Total number of donors per group is indicated in parentheses. LC–MS signals were expressed as fold change relative to the upper limit of normal (ULN, grey line). Violin plots are superimposed to visualise the distribution of all samples including those with low (<1/32 ULN) or without peptide detection

responses to CCNI-ED10 (Fig. 2b). Thus, our data demonstrated that edited peptides can function as antigens to stimulate T-cell responses in tumour tissues.

**Edited peptide-specific T cells mediate tumour cell killing.** The identification of CCNI-ED as an antigen able to activate human T cells prompted us to determine the function of CCNI-ED specific T cells in mediating tumour cell killing. For this experiment, we employed the HLA-A*02:01-expressing lymphoblast cell line T2 which is lacking expression of the transporter associated with antigen processing (TAP) and thus incapable of presenting endogenous peptides[30]. We pulsed edited and wild-type CCNI peptides onto T2 cells, co-cultured with TILs in different ratios, and then measured T-cell mediated target cell death based on anti-caspase3 staining and subsequent flow cytometric quantification[31]. A pulse of CCNI-ED10, but not CCNI-WT10, to T2 cells facilitated target killing of both TIL2661 and TIL2678 (Fig. 2c and Supplementary Fig. 2), suggesting that CCNI-ED10 was a target of cytotoxic T lymphocyte (CTL) activity. To confirm whether CCNI-ED could also mediate target killing activity under natural antigen processing conditions, we cloned the cDNA

encoded wildtype or edited *CCNI* full-length protein and transiently transfected the HLA-A*02:01-expressing human embryonic kidney cell line (HEK 293-A2, ATCC). Over-expression of edited protein in 293-A2 was associated with profound cytotoxic activity of TIL2678, whereas expression of wildtype cDNA or empty vector resulted in background levels of cytotoxic activity (Fig. 2d and Supplementary Fig. 3).

**Edited RNA as a suitable surrogate for edited peptides.** Measurement of RNA editing on peptide-level by direct immunopeptidome analysis is cost and labour-intense thus to enable further characterisation, analysis on mRNA level is preferred. This requires confirmation that edited mRNA levels indeed correlate with the number of edited peptides bound to HLA. Making use of the quantitative HLA peptidomics data shown in Fig. 1b, we integrated the peptidome data with corresponding RNA-seq measurements of matched samples to correlate the peptide abundance with the expression of edited *CCNI*. Our results revealed a weak correlation for CCNI-ED9 ($R = 0.33$, 95% confidence interval $CI = 0.03$–$057$, Supplementary Fig. 4a), but a

**Table 1 List of HLA-bound edited peptides (ED) and their non-edited (WT) counterparts as identified by MS-based immunopeptidomics from primary human tissue**

| Editing site (hg19 coordinates) | Gene | Edited peptide (sequence) | Wildtype peptide (sequence) | HLA restriction |
|---|---|---|---|---|
| CCNI R75G (chr4:77,979,680) | Cyclin I | CCNI-ED9 (LLDGFLATV) | CCNI-WT9 (LLDRFLATV) | A*02:01 |
| CCNI R75G (chr4:77,979,680) | Cyclin I | CCNI-ED10 (SLLDGFLATV) | CCNI-WT10 (SLLDRFLATV) | A*02:01 |
| COPA I164V (chr1:160,302,244) | Coatomer subunit alpha | COPA-ED10 (RVWDVSGLRK) | COPA-WT10 (RVWDISGLRK) | A*03:01 |
| COPA I164V (chr1:160,302,244) | Coatomer subunit alpha | COPA-ED11 (RVWDVSGLRKK) | COPA-WT11 (RVWDISGLRKK) | A*03:01 |
| CDK13 Q35R (chr7:39,990,344) | Cyclin Dependent Kinase 13 | CDK13-ED (SPRQPPLLL) | CDK13-WT (SPQQPPLLL) | B*07:02 |

Amino acids derived from RNA editing are underlined

strong correlation for CCNI-ED10 ($R = 0.67$, $CI = 0.45–0.81$, Supplementary Fig. 4b).

In order to determine overall editing on peptide level we performed absolute quantitation which allows to combine both length variants quantitatively. For CCNI-ED10 and CCNI-ED9, we measured an average number of 37 and 32 edited peptide copies per cell, respectively, while 261 and 336 copies of non-edited peptide were detected for CCNI-WT10 and CCNI-WT9, respectively. For this purpose, a set of 8 samples was successfully analysed by parallel reaction monitoring (PRM) LC–MS and RNA-seq targeting CCNI peptides and corresponding mRNA, respectively. We observed a very strong correlation ($R = 0.96$, $CI = 0.80–0.99$, Fig. 3a) between both levels which allowed us to investigate the prevalence of over-editing on mRNA level in The Cancer Genome Atlas (TCGA) for external validation on a larger cohort. Over-editing analysis was done in the same fashion as on peptide level by determining the ULN and the fold change of edited transcript relative to the ULN (Supplementary Fig. 5a). For the TCGA studies matching the cancer types investigated on HLA peptidome level, we observed a prevalence of over-editing of 3.4% ($n = 210/6106$) with the highest prevalence for ovarian cancer (OV, 23%, $n = 67/293$), breast cancer (BRCA, 8%, 87/1095) and kidney cancer (KIRC, 6%, $n = 26/448$).

**CTL activity depends on *ADAR1* mediated target editing.** Having established the correlation between edited peptide and mRNA, we used TCGA tumour data to extend our mechanistic understanding of *CCNI* editing. It has been suggested that *CCNI* R75G is primarily edited by *ADAR1*[12,32]. To investigate the situation in tumour tissues, we correlated the expression of *ADAR* transcripts with expression of edited *CCNI*. This analysis revealed higher correlation for *ADAR1* ($R = 0.48$, $CI = 0.47–0.50$, Supplementary Fig. 5b) than for *ADAR2* ($R = 0.28$, $CI = 0.26–0.30$) or *ADAR3* ($R = 0.07$, $CI = 0.05–0.10$) suggesting *ADAR1* as most likely enzyme responsible for catalysing the editing. This is also supported by associating the quantitative HLA-A*02 peptidome data (Fig. 1a) directly with the corresponding mRNA expression measurements. Logistic regression modelling of the detection of CCNI-ED10 by *ADAR1* mRNA expression showed significant association for *ADAR1* (Odds ratio $OR = 30.8$, $p < 0.001$) in contrast to *ADAR2* ($OR = 2.1$, $p = 0.157$) and *ADAR3* ($OR = 0.8$, $p = 0.572$). To provide additional experimental evidence, we transfected HEK 293 with *ADAR1* and *ADAR2*. This resulted in elevated *CCNI* editing only for *ADAR1* (Fig. 3b, Supplementary Fig. 6), supporting a causal relationship.

To investigate the correlation of CCNI-ED10-specific T cell killing activity to the endogenous target editing level, we generated CCNI-ED10-specific effector T cells (Ted10) from HLA-A*02:01-expressing normal PBMCs derived from two healthy donors using an established method[33]. Importantly, the Ted10 cells were potently activated by 293-A2 cells pulsed with CCNI-ED10 or transfected with the edited gene, as demonstrated in Fig. 3c and Supplementary Fig. 7. Furthermore, the Ted10 cells

displayed substantially elevated killing activity towards 293-A2 cells over-expressing the edited gene compared to background killing activity towards control cells transfected with the wildtype or empty vector (Fig. 3d). Activation of TIL2661 and TIL2559 (Fig. 2a) by CCNI-ED10 suggested that this epitope was presented to TIL2661 and TIL2559 endogenously by autologous tumours resulting in the response to CCNI-ED10 in the absence of in vitro education. Based on the established correlation between edited mRNA and peptide, we performed RNA-sequencing analyses to detect *CCNI* mRNA editing in melanoma cell lines, including mel-2661, mel-2559 and mel-2400 derived from the patients used for generating TIL2661 and TIL2559. Indeed, we were able to detect endogenously edited *CCNI* mRNA (Fig. 3e) except for mel-2559, which was derived from the same patient as mel-2400 yet from a different tumour sample. *ADAR1* mRNA was 40 times lower in mel-2559 ($\Delta C_t = 0.073$) compared to mel-2400 ($\Delta C_t = 2.93$) based on qPCR measurements. We next examined the functional significance of endogenous CCNI-ED10 in T-cell activation using Ted10. IFNγ ELISPOT assays detected strong reactivity of Ted10 to mel-2400 and mel-2661 as well as another CCNI-editing positive tumour line, mel-2391, whereas Ted10 cells displayed only a background level response to the CCNI-editing negative mel-2559. Furthermore, Ted10 did not react to HLA-A*02:01 negative mel-2686 and mel-2357 despite their expression of edited mRNA (Fig. 3e), confirming HLA-restricted and antigen-specific T-cell activation. Parallel CTL assays showed that Ted10 displayed strong killing activity towards the CCNI-editing positive mel-2400 but had almost no activity against the autologous CCNI-editing negative mel-2559 (Fig. 3f). Furthermore, knockdown of *ADAR1* in mel-2400 greatly reduced its ability to stimulate Ted10 to produce IFNγ (Supplementary Fig. 8).

**Discussion**
We hereby present for the first time that HLA-bound peptides derived from RNA editing function as tumour antigens to elicit immune responses. We were able to confirm the existence of five peptides derived from three editing sites out of 1369 sites investigated. While this might seem to represent a low proportion, this gap between the number of RNA events observed and HLA-bound peptides detected is in the range of other proteogenomics studies which also report less than 1% of the genomic sites to be presented on peptide level. For instance, Granados et al.[34] identified 26 peptides out of 4833 SNPs (0.54%), Yadav et al.[35] identified 7 out of 1357 expressed somatic tumour mutations (0.52%) and Bassani-Sternberg et al.[36] identified 11 out of 3487 mutations (0.32%). While LC–MS sensitivity can be considered as one factor, reverse immunology identifications were also in this range with 18 confirmed mutations out of 1452 as reported by Tran et al.[37]. More importantly this gap is influenced by biological factors including mRNA stability, translational regulation, protein turnover, proteasome processing, cytosolic peptidases, TAP and binding affinity to HLA[38].

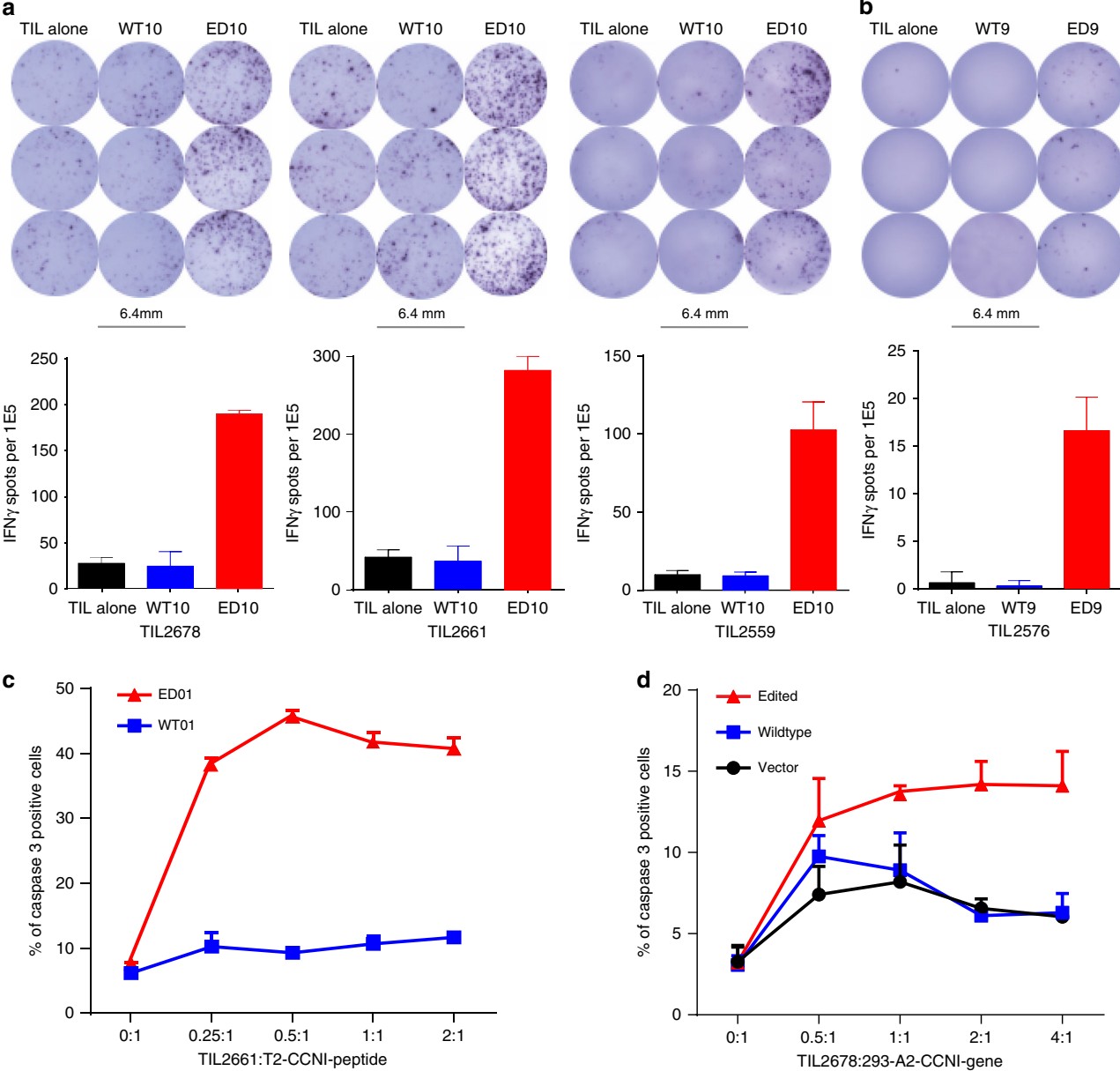

**Fig. 2** Response of TILs to CCNI peptides. **a** ELISPOT (upper) and summary graphs (lower) showing IFNγ production (arithmetic mean and s.e.m., n = 3) by 3 melanoma TILs following incubation with CCNI-ED10 peptide. Only background level of IFNγ-producing TILs were detected when incubated alone or together with CCNI-WT10. **b** TIL2576 weakly reacted to CCNI-ED9. **c** Caspase3 based Cytotoxic T Lymphocyte (CTL) killing assay showing TIL2661 mediated killing of T2 cells pulsed with CCNI-ED10 or CCNI-WT. **d** Overexpression of edited but not wildtype CCNI gene in 293-A2 cells enhanced TIL2678's CTL killing activity. The error bar represents the standard error of the mean (s.e.m.) of the three replicates

For in-depth characterisation we focused on the *CCNI R75G* editing site that presented two over-lapping HLA-A*02 ligands, CCNI-ED10 and CCNI-ED9. We quantified both peptides on a comprehensive set of tumour and normal tissues to assess potential tumour association. The edited peptides showed abundances elevated beyond normal levels ("over-editing") in 7% of all tumours, which is a prevalence comparable to the most frequent neoantigen epitopes like PIK3CA H1047R, which is expressed in 4% of all tumours[39]. The tumour indications with the most frequent over-editing were ovarian cancer, melanoma and breast cancer showing a prevalence comparable to frequent neoantigen epitopes like KRAS G12D which is present in 33% of all pancreatic cancer patients[39,40]. While over-editing cannot be considered tumour-specific due to presentation of edited peptides on healthy tissues, it contributes to the class of over-expressed and

thus tumour-associated shared self-antigens[23]. As shown e.g. for MUC1 and other over-expressed self-antigens, central tolerance can be incomplete or absent for self-peptides that are associated with tumours due to over-expression, tissue- or germline-specific expression[8,41]. While this antigen class sets higher standards for clinical development with regard to definition of an appropriate therapeutic window and establishing safety, it allows for immunotherapeutic approaches in indications with low mutational load especially when multiple peptide targets are combined to a warehouse for active personalisation[42].

To confirm the prevalence of over-editing on a larger scale, we investigated *CCNI* editing in TCGA on mRNA level. While an overall correlation between transcript levels and number of detectable HLA peptides has been shown[43], the peptide-specific abundance levels correlate only for a fraction of HLA peptides

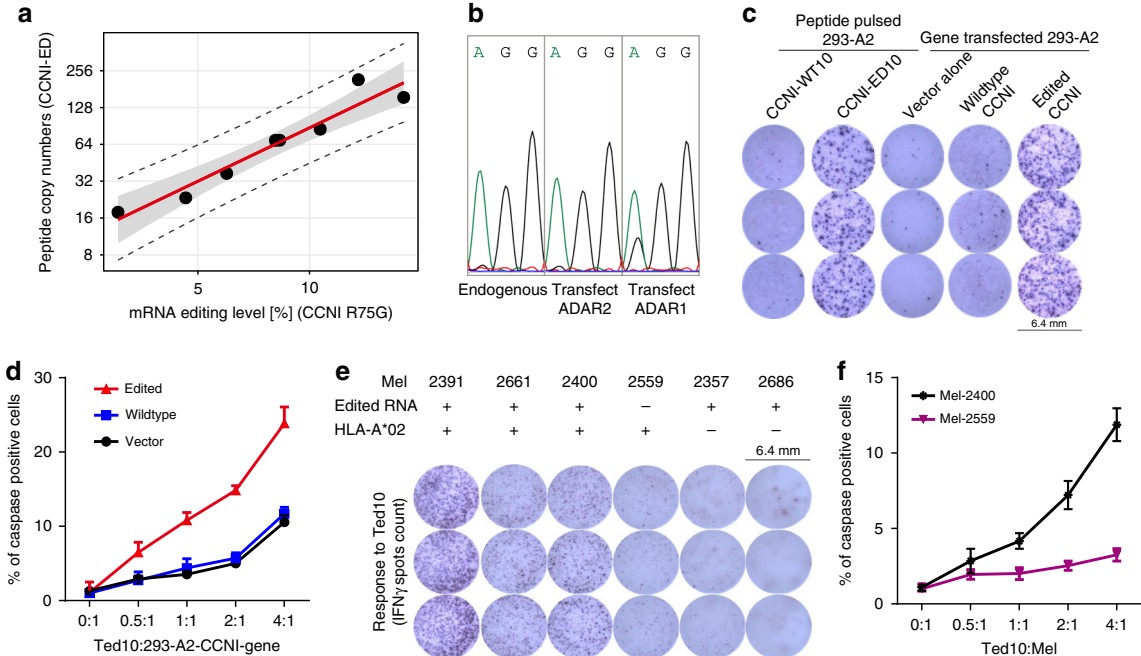

**Fig. 3** Edited CCNI peptide level is correlated with its biological function. Correlation of CCNI peptide levels with *CCNI* and *ADAR* mRNA expression as well as Ted10 activation and Ted10 mediated tumour target killing. **a** Correlation between the number of edited CCNI peptide copies per cell determined by the AbsQuant® method and mRNA editing levels determined by targeted RNA-seq for *CCNI R75G*. The scatterplot (n = 8) includes the regression curve (red line) as well as the 95% confidence interval (grey band) and 95% prediction interval (dashed lines). **b** *CCNI-R75G* is edited by *ADAR1*. HEK 293 stably expressing *CCNI* wildtype gene was transfected with empty vector or expression vectors of *ADAR1* or *ADAR2*. *CCNI* editing was measured by RT-PCR and followed by sequencing. The double peaks indicate nucleotide A to G conversion and the height of peaks reflect the level of editing. **c** ELISPOT assay showing IFNγ production by Ted10 incubated with peptide-pulsed or *CCNI*-transfected 293-A2 cells. **d** CTL killing assay showing that over-expression of edited *CCNI* gene increases the sensitivity of Ted10 mediated 293-A2 target killing (n = 3), summarised as mean ± s.e.m. per titration. **e** IFNγ ELISPOT assay showing recognition of endogenous CCNI-ED antigen by Ted10. Mel-2391, mel-2400 and mel-2661 expressing both edited *CCNI* mRNA and HLA-A*02:01 are highly reactive to Ted10. Mel-2559, which was derived from the same patient as mel-2400 but does not have detectable edited *CCNI* mRNA, only reacted at background levels to Ted10. Mel-2357 and mel-2686, which express edited *CCNI* mRNA but do not express HLA-A*02:01, have no response to Ted10. **f**, Ted10 mediated target killing following incubation with mel-2400 and mel-2559 measured by caspase-3-based CTL killing assay (summarised as mean ± s.e.m. of the three triplicates)

with their corresponding mRNA[44,45]. Thus, confirmation of correlation between number of edited peptide copies and edited transcripts is necessary. We were able to show that for edited CCNI peptides the mRNA can be used as a surrogate measurement for peptide levels, and based on this we determined prevalence estimates on the TCGA data set and showed that they reproduce peptide-level estimates. While TCGA enables a comprehensive assessment of tumour and autologous normal tissues, the included normals are underrepresented and adjacent to tumour tissue. Therefore, for a comprehensive safety assessment further studies incorporating healthy donors will be necessary.

To investigate tumour association on cellular level, we screened melanoma TILs for edited peptide-specific T cells and demonstrate that peptides derived from RNA editing are immunogenic. We have found T-cell populations from TILs that respond to both CCNI-ED 10 and CCNI-ED 9 and showed that these TILs are activated in vitro by the edited peptides and display specific cytotoxicity toward target cells expressing endogenous edited protein. Although this study only investigated the CCNI-ED peptides in depth, our finding that 3 out of 15 TILs positively reacted to CCNI-ED 10, and 1 out of 15 TILs reacted to CCNI-ED 9, suggests the possibility that edited peptides serve as a source of antigens that activate tumour-specific T cells which infiltrate into tumour site to mediate antitumor immunity. The editing of *CCNI* mRNA is mediated by *ADAR1*, which is the major enzyme in RNA editing[12]. Our data further demonstrate that both the level of *CCNI* RNA editing in target cells and the activation of CCNI-ED-specific T cells depend on *ADAR1* expression.

In summary, for the first time we identified RNA editing products, particularly the CCNI-ED10 peptide, as immunogenic epitopes that are presented on HLA and are able to stimulate T-cell responses. We characterised the epitope's target potential by MS-based immunopeptidomics showing a quantitative profile of an RNA-edited HLA-bound peptide on a comprehensive panel of primary human A*02-positive tissues as well as direct correlation between peptide level and mRNA. Edited peptides were found on healthy and tumour tissues showing over-editing for a subpopulation of tumours. Thus, over-edited peptides present an additional class of tumour-associated self-antigens that might provide a therapeutic window for immunotherapies. While safety needs to be addressed in future studies to exclude on-target toxicity, we show that T cells with cytotoxic reactivity against edited peptides are physiologically present in cancer tissue and thus in patients without evidence of severe side effects. The shared nature of these antigens would suggest new opportunities for immunotherapies in the treatment of cancer and immunological disorders.

## Methods

**Peptide isolation and mass spectrometry**. To allow discovery and selection of novel HLA peptides as targets for immunotherapy, we acquired immunopeptidomes together with corresponding transcriptomes and HLA genotypes for 1514 primary human tissue samples extracted post mortem or surgically from 850 patients with cancer or benign neoplasms and 269 healthy tissue donors after written informed consent. The resulting sample set of 616 normal and 898 cancer samples covered 35 different organs and 23 tumour types with at least 5 donors per group and a median group size of 16 donors. Samples were snap-frozen in liquid nitrogen and stored until

isolation at −80 °C. After tissue homogenisation and lysis, peptide-MHC complexes were isolated by immunoprecipitation using class I specific antibodies coupled to CNBr-activated Sepharose resin (GE Healthcare Europe, Freiburg, Germany). Depending on the donor's HLA type, the following antibodies were used according to Falk et al.[46]: w6/32 for pan-class I, BB7.2 for HLA-A*02, GAP-A3 for HLA-A*03 and B1.23.2 for HLA-B/C (Department of Immunology, University of Tübingen, Germany). Peptides were eluted from antibody-resin by acid treatment and purified by ultrafiltration. For further separation, reversed-phase chromatography (nanoAcquity UPLC system, Waters, Milford, MA) was used eluting with an ACQUITY UPLC BEH C18 column (75 μm × 250 mm, Waters, Milford, MA) at a 190 min gradient ranging from 1 to 34.5% ACN. Eluted peptides were analysed by data-dependent acquisition (DDA) in an Orbitrap mass spectrometer (Thermo Fisher Scientific, Waltham, MA) equipped with a nano electrospray ionisation (ESI) source. A total of 7825 runs was acquired in profile mode covering most samples with five replicate injections making use of different mass analysers in low- (TOP3, ion trap acquiring top 3 precursors) and high-resolution mode (TOP5, Orbitrap acquiring top 5 precursors, $R = 7500$), as well as different fragmentations using collision-induced dissociation (CID) and higher-energy collisional dissociation (HCD). Survey scans were acquired with high mass accuracy in the Orbitrap ($R = 30,000$ for TOP3, $R = 60,000$ for TOP5). Mass range for selection of doubly charged precursors was 400–750 m/z and 800–1500 m/z for singly charged precursors. Spectra were extracted and centroided using Proteome Discoverer 1.4 (Thermo Fisher Scientific, Waltham, MA).

**RNA isolation and sequencing.** Immunopeptidome measurements were accompanied by paired transcriptome analysis for a subset of 276 samples by isolating total RNA using TRIzol® (Invitrogen, Karlsruhe, Germany) followed by a purification with the RNeasy mini kit (QIAGEN, Hilden, Germany) according to the manufacturer's protocol. RNA sequencing and expression quantification was performed by CeGaT (Tübingen, Germany). Briefly, 1–2 μg total RNA were used as starting material for library preparation performed according to the Illumina® protocol (TruSeq Stranded mRNA Library Prep Kit). Sequencing was performed on an Illumina® HiSeq® 2500 machine. For all experiments, a strand-specific protocol was used to generate single-end reads of a length of 50 nucleotides. The minimum number of reads was 43,700,000 per sample. The quality of the sequencing process was monitored using PhiX spike-ins. We performed DESeq[47] to determine normalisation factors to allow inter-sample read count comparisons.

For eight samples with detectable copy numbers of edited peptide and remaining mRNA available, expression of edited CCNI was measured by CeGaT (Tübingen, Germany) using targeted RNA-seq. Briefly, 100 ng total RNA were used and amplified specifically for CCNI R75G using the primers 5′-GATGTGGAA AGTGAATGTGCG-3′ (forward) and 5′-TTTGGATGAGCCTTTACGGTAG-3′ (reverse). Library preparation was performed according to Illumina® protocol (Nextera XT Index PCR System) followed by sequencing on an HiSeq® 2500 generating about 10 million paired-end reads with length of 2 × 100 nucleotides.

**HLA typing.** To experimentally assess the HLA restriction of a given peptide, DNA of donors was isolated from tissue or whole blood using the QIAamp® DNA Mini Kit (Qiagen) or the QIAamp® DNA Blood Mini Kit (Qiagen), respectively. The QIAamp® Investigator Kit (Qiagen) has been used to isolate DNA from very limited amounts of tissue. HLA genotyping for HLA-A*02 was performed by PCR and subsequent agarose gel electrophoresis using the Ambisolv® Primer Mix PM002 (Life Technologies) and recombinant Taq polymerase (Life Technologies). Fine typing of the HLA-A and -B loci were performed by Sanger sequencing using the SeCore® Sequencing Kits (Invitrogen/Life Technologies). SeCore® Custom GSSP Kits (Invitrogen/Life Technologies) were used to resolve ambiguities if necessary. Samples were sequenced on an ABI-3100 sequencer (Applied Biosystems/Thermo Fisher Scientific) at CeGaT (Tübingen, Germany) and results were evaluated using the uTYPE™ software (Invitrogen/Life Technologies).

**Proteogenomics peptide identification.** The RNA editing sites referenced during the study are available in a public repository from the RADAR website (http://rnaedit.com)[27]. RADAR version 2 contained 2,576,459 entries that were downloaded and annotated using ANNOVAR based on the RefSeq annotations hg19_refGeneMrna.fa and hg19_refGene.txt[48]. Filtering for non-synonymous events resulted in 1,369 RNA editing sites. Amino acid sequences were inferred using the R package sapFinder[49]. Due to different protein isoforms, the editing sites correspond to 2516 entries which correspond to 1387 unique different candidate peptides (21mers) with up to ten amino acids before and after the editing site (Supplementary Data). The editing peptide database was concatenated with the reference proteome (UniProt 2016-04-13)[50] and a reversed version thereof for MS/MS database search using Comet (v2016.01 rev.2)[51]. The search was performed with the following parameters: peptide length 8–12 AAs, mass range 700–1500 Da, non-specific enzymatic digestion, precursor mass tolerance 3 ppm, 0.02 Da bin size for high resolution (Orbitrap) spectra and 1 Da for low resolution (Ion trap) spectra, and methionine oxidation as variable modification. The Comet search results were then analysed by PeptideProphet (TPP v5.0.0)[52] that estimates a probability score for each Peptide-spectrum-match (PSM) with assistance of decoy hit scores. The PSMs from all samples were further grouped into individual peptide ions (distinct peptide sequence, modification, and charge state) and the best

probability score was taken to represent the final score for each peptide ion. False discovery rate (FDR) for each peptide ion was then estimated by target-decoy approach[53]. Supplementary Table 1 lists all RNA editing sites found at 1% FDR as well as all peptide ions at 5% FDR that cover those sites including different length variants and charge states. The identified peptide ions were inspected for MS/MS matching quality, HLA restriction, and RNA-seq support. MS/MS matching quality was assessed by inspecting the matched fragment ion coverage and coverage of dominant peaks. In case of questionable coverage, alternative sequences suggested by denovo identification using PepNovo + (2010-11-17)[54] were considered. HLA restriction of an identified edited peptide was determined by comparing the potential MHC peptide binding motif[55] with the experimental HLA typing of the corresponding sample and the specificity of the antibody used for immunoprecipitation (see Supplementary Table 1). RNA-seq experiments of mRNA extracted from the same sample as the peptide eluates were analysed using samtools 0.1.19 to find supporting reads[56].

**Peptide sequence validation.** To experimentally validate the edited peptide sequence, peptides were synthesised on an automated Prelude® peptide synthesiser (Protein Technologies Inc., Tucson, AZ) using solid phase peptide synthesis (SPPS) and Fmoc-chemistry. C[13]/N[15]-labelled Leucine (Cambridge Isotope Laboratories Inc., Tewksbury, MA) was used to isotopically label the peptide resulting in a mass shift of 7.017 Da. The isotope-labelled peptides were spiked into retention vials of the original sample and analysed by LC–MS. The heavy labelled peptide variant creates a reference signal that has the same chromatographic properties yet does not interfere with the native mass signal. To unambiguously validate the sequence identity of edited peptides we compared the elution times and the fragmentation pattern of labelled and native peptide signals (Supplementary Fig. 1). To detect peptides with maximum sensitivity, the MS/MS spectra were acquired by data independent mode (DIA) restricting to labelled and native peptide masses. Fragmentation patterns were generated using XCalibur 3.0.63 (Thermo Fisher Scientific) and elution profiles were extracted using Skyline 3.6.0[57].

**Relative quantitation of peptides.** For direct quantitation of CCNI peptides on peptide presentation level, 421 normal and 504 tumour samples were chosen based on the following criteria. The donor tested positive for A*02 based on HLA typing and the BB7.2 immunoprecipitation resulted in at least 100 identified peptides based on at least four evaluable technical replicates. LC-MS peptide signal features were extracted by SuperHirn v1.0[58] to determine peak areas for extracted ion chromatograms (XIC) allowing MS1-based relative quantitation. After charge state deconvolution with OpenMS Decharger 1.6[59], LC–MS features were assigned to identified MS/MS spectra. To allow maximum sensitivity, identification was based on a spectral library approach using 5% posterior error probability (PEP) fitted by R mixtools 1.1.0[60] on cross-correlation scores between manually confirmed reference spectra and all available MS/MS within a 10ppm precursor range. Peptide abundance levels per sample were determined by median total-area of the replicates. The total-area was defined as the sum of the normalised XIC areas of all observed charge states. Systematic bias was rectified by central tendency normalisation[58] to account for differences in MHC expression and technical variations.

**Statistical analysis of immunopeptidome data.** Statistics and figures were generated using R 3.4.2 and ggplot2 2.2.1. Tumour association of CCNI peptides was analysed by comparison against the normal reference range. Peptide abundances of normal samples were grouped according to organ and the 95th percentile (P95) was determined for each organ. The upper limit of normal (ULN) was defined as maximum P95 over all healthy samples and tumour samples above ULN were classified as over-edited. For visualisation, peptide abundances were presented as fold change with respect to ULN and grouped according to tumour type or healthy organ. Every group consisted of more than 5 samples. Normal samples from cartilage, bile duct, eye, thymus, central nerve, spinal cord and pleura did not meet this requirement and were grouped into the category other. The distribution of fold changes for each group was estimated as violin plots. Samples with values below 1/32 ULN or without detection of the peptide were set to 1/32 ULN.

Association between label-free LC-MS and corresponding RNA-seq data was analysed between CCNI peptides and CCNI as well as ADAR gene expression. To investigate if mRNA levels are predictive for peptide presentation levels, the correlation between peptide quantitation and normalised read count for CCNI R75G edited reads was analysed by Pearson's correlation coefficient based on all samples with pairwise complete observations ($n_{ED9} = 44$, $n_{ED10} = 39$). Association with ADAR was analysed with logistic regression modelling for all pairwise measurements. The likelihood of peptide detection was modelled by log-transformed reads per kilobase per million mapped reads (RPKM) for ADAR1-3.

**Absolute quantitation of peptides.** For a set of 22 samples absolute copy numbers per cell where measured using the AbsQuant® method. In brief, copy numbers of CCNI peptides were calculated using number of cells within the investigated tissue and total amount of the isolated peptide. Hereby, both parameters were determined experimentally. The isolation efficiency was assumed to be 100%. For three samples determination of cell count was not possible whereas one sample had no mRNA left for targeted RNA-seq analysis. Ten samples were not evaluable due

to LC-MS issues in detection of one or both CCNI-ED peptides. This resulted in a set of eight samples with evaluable copy numbers.

The number of cells was determined based on quantitation of DNA content in the investigated human tissue sample. Therefore, DNA was isolated using QIAamp® DNA Mini Kit (QIAGEN, Hilden, Germany) from lysate aliquot which was sampled during the isolation of HLA ligands from primary tissue. The DNA yield was quantified using Qubit™ dsDNA HS Assay Kit (Applied Biosystems/Thermo Fisher Scientific) and the number of cells was interpolated from DNA content using a standard curve derived from peripheral blood mononuclear cells (PBMCs).

For absolute quantitation of CCNI peptides, a series of nanoLC-MS/MS measurements was performed on an Orbitrap mass spectrometer (Thermo Fisher Scientific, Waltham, MA) using parallel reaction monitoring (PRM). Two differently isotopically labelled CCNI peptide equivalents were synthesised as described above. One of the isotopically labelled equivalents was used as an absolute quantity reference and was spiked into retention vials of each human tissue sample which was used for absolute quantitation of CCNI peptides. The other isotopically labelled equivalent was used to generate the peptide-specific standard curve. Thereby, one of the isotopically labelled equivalents was titrated and the other one was used as mentioned before as an absolute quantity reference. The MS/MS spectra were acquired by data independent mode (DIA) restricting to labelled peptide masses by the analysis of standard curves and labelled and native peptide masses by the analysis of primary tissue samples. The MS/MS signals of selected fragment ions were extracted using *Skyline 3.6.0*[57] and interpolated in absolute peptide amount using peptide-specific standard curves. The number of edited copies per cell was defined as sum of copies for CCNI-ED9 and CCNI-ED10. Values below limit of detection (LOD) or lower limit of quantitation (LLOQ) were imputed with the respective thresholds.

**TCGA data analysis**. We downloaded 9233 RNA-seq bam files of normal and tumour TCGA samples[61]. The number of edited and total reads at chr4:77,979,680 was extracted as well as gene expression of *CCNI* and *ADAR1* to *ADAR3* as transcripts per kilobase million (TPM). Reads were extracted using samtools[56] and filtered according to base quality ≥ 20 and mapping quality ≥ 20. Pearson's correlation coefficients between *CCNI* and ADARs were determined after log-transformation for all tumour samples ($n = 8522$) with pairwise complete non-zero values ($n = 8241$ for *ADAR1/2*, $n = 6666$ for *ADAR3*). Over-editing analysis was done analogously to the peptide data. For estimation of the reference range, autologous normal samples from TCGA were grouped according to organ if more than 5 samples existed. Normal samples from brain, pancreas, skin, thymus and soft tissue did not meet this requirement and were grouped into the category *other*. Tumour samples were restricted to studies with patient populations comparable to those used for immunopeptidome quantitation covering bladder urothelial carcinoma (BLCA), glioblastoma multiforme (GBM), hepatocellular carcinoma (LIHC), ovarian serous cystadenocarcinoma (OV), acute myeloid leukaemia (LAML), oesophageal carcinoma (ESCA), stomach adenocarcinoma (STAD), cholangiocarcinoma (CHOL), colon adenocarcinoma (COAD), rectum adenocarcinoma (READ), pancreatic adenocarcinoma (PAAD), lung adenocarcinoma (LUAD), kidney renal clear cell carcinoma (KIRC), lung squamous cell carcinoma (LUSC), prostate adenocarcinoma (PRAD), lymphoid neoplasm diffuse large B-cell lymphoma (DLBC), skin cutaneous melanoma (SKCM), uterine corpus endometrial carcinoma (UCEC), head and neck squamous cell carcinoma (HNSC) and breast invasive carcinoma (BRCA).

**Generation of TILs and tumour cell lines**. The TILs and tumour cell lines used for experimental validation were derived from residual tumour tissue obtained from metastatic melanoma patients enroled on an adoptive cell therapy clinical trial using TILs at the University of Texas MD Anderson Cancer Center (Institutional review board (IRB)-approved protocol # 2004-0069, NCT00338377). All patients had granted a written informed consent.

Melanoma TILs were generated according to Dudley et al.[4,62]. Briefly, melanoma tumour samples were either cut into 1–3 mm² fragments and put in culture in a tissue-treated 24-well plate, in complete TIL media (TIL-CM) consisting of RPMI 1640 (Gibco, 61871), 10% human AB serum (GEMINI,100–512), 0.1% 2-mercaptoethanol (Gibco, 21985023), 1% HEPES (Corning, 25–060), 1% sodium pyruvate (Invitrogen/Life Technologies, 11360-070), 1% Glutomax (Gibco, 35050061), 1% PenStrep (ThermoFisher,15070063) plus 6000 U per mL of human IL-2 or put in culture after the tumour samples were enzymatically digested by collagenase for 1 h at 37 °C followed by centrifugation using a multi-layer Ficoll gradient (100 and 75%, Ficoll-Paque PLUS,GE, 17-1440-02, at $800 \times g$ for 30 min) where the 100% layer was collected for TILs. Every 3 days, half of the medium was replaced with fresh medium and the TIL culture was split to keep the cells at a density of $1 \times 10^6$ per mL. TILs were expanded between 2 and 5 weeks, depending on the TIL lines. To increase the number of TILs available for experiments, the lines were further expanded using the rapid expansion protocol (REP)[4,62]. In brief, $1.5 \times 10^5$ primary TILs generated above were cultured with $27 \times 10^6$ feeder cells together with 0.6 mg soluble anti-CD3 monoclonal antibody (OKT3 clone, Muronomab—Abbott Labs). The feeder cells were peripheral blood mononuclear (PBMC) cells mixed from at least 5 healthy donors and irradiated at 5000 cGy for 20 min prior to culture in order to prevent

their proliferation during the REP. In total 6000 U per mL IL-2 was added at the second day and half of the medium was recovered and replaced with fresh medium containing 50% of TIL-CM and 50% of AIM-V medium (Invitrogen, 12055-083) every 3 days to keep TIL density between 0.5 and $2 \times 10^6$ per mL. The cultured TILs were harvested at day 14 for functional analysis or frozen in human serum with 10% DMSO (Thermo Fisher, 67-68-5). Autologous tumour cell lines were also established from enzymatically digested tumours followed by multi-layer Ficoll gradient where the 75% layer was collected and put in culture in complete tumour media (RPMI 1640, Gibco, 61871) containing 10% FBS, 1% HEPES (Corning, 25–060), 1% sodium pyruvate (Invitrogen/Life Technologies, 11360-070), 1% insulin/selenium/transferrin (Gibco, 51300), 0.2% MycoZap-PR (Lonza, VZA-2011). All tumour samples were HLA typed at the HLA-A locus in the MD Anderson HLA Typing Laboratory. The cell lines were routinely tested for mycoplasma contamination (Lonza, LT07-418).

**ELISPOT assay**. IFNγ Enzyme-linked immunospot (ELISPOT) assay was performed to detect T-cell responses. MultiScreen 96 well filter plates (Millipore, MAHAS4510) were coated over night at 4 °C with 75 µL per well of 5 ng per mL anti-human IFNγ capture antibody (Mabtech AB, 3420-3-1000). TILs or CCNI-ED10 specific T cells (Ted10) were thawed and cultured with 1000 U of human IL-2 per mL overnight. The next day, before performing ELISPOT assay, T cells were starved with IL-2 free medium for 6 h. T cells were then added into plates in triplicates at $2 \times 10^5$ cells per well (for TIL) and $0.4 \times 10^5$ per well (for Ted10) or as indicated in each experiment with culture medium either alone or supplemented with peptides (10 µM final concentration), peptide-pulsed T2 ($1 \times 10^5$ per well), 293-A2 cells ($1 \times 10^5$ per well) or melanoma cell lines ($1 \times 10^5$ per well). After 18 h of cultivation at 37 °C and 5% CO₂, the plates were incubated with 1 ng per mL of Biotinylated anti-human IFNγ monoclonal antibody (Mabtech, 3420-6-1000) for one hour, stained with ExtrAvidine®-Alkaline phosphatase (Sigma-Aldrich, E2636) and IFNγ positive spots were detected with BCIP/NBT Membrane Alkaline Phosphatase Substrate (Sigma, 11697471001). Plates were scanned and counted using the ImmunoSpot® ELISPOT analyser (Shaker Heights, OH) to determine the number of spots per well.

**Peptides and tetramers**. Synthetic peptides used in this study were obtained from Genemed Synthesis, Inc (San Antonio, TX) or were synthesised as described above (Immatics®, Tübingen, Germany). All peptides were purified by HPLC to provide a homogeneity of >95%. The tetramer was made by Protein Chemistry Core-MHC Tetramer Lab in Baylor College of Medicine (Houston TX).

**Peptides pulsing**. In total $5 \times 10^6$ T2[30] or 293-A2 cells in 1 mL of PBS supplemented with 1% FBS (foetal bovine serum) were incubated with synthetic edited and non-edited peptides at 10 µM final concentration for 2 h at 37 °C incubator and washed once with T-cell medium before being subjected to ELISPOT or Caspase-3-based CTL killing assay.

**In vitro generation of peptide-specific T cells**. Peptide-specific T cells were generated from normal donor peripheral blood mononuclear cells (PBMCs) as described by Li et al.[33], and leukapheresis were purchased from Key Biologics (Memphis, TN). The adherent monocytes from PBMC were cultured for one week with 800 U per mL of recombinant human GM-CSF (Thermo Fisher, 215-GM) and 500 U per mL of recombinant human IL-4 (R and D, 204-IL-050) to generate dendritic cells (DCs) and then treated for 24 h with 10 ng per mL of recombinant human TNFα (R and D, 210-TA), 2 ng per mL of recombinant human IL-1β (R and D, 201-LB-005), and 1000 U per mL of recombinant human IL-6 (R and D. 206-IL-010) plus 1000 ng per mL of Prostagladin E2 (MP Biomedicals, 219457601) to induce DC maturation. Usually, $1.8 \times 10^6$ matured DCs were then pulsed with 10 µM peptides for 4 h at room temperature in PBS supplemented with 1% human serum. Peptide pulsed DCs were then irradiated for 20 min at 5000 rad. Autologous PBMC were then mixed with DCs at 35:1 ratio and cultured in T-cell medium supplemented with 30 ng per mL of recombinant human IL-7 (R and D, 1 207-IL-005) and 5 ng per mL of recombinant human IL-21 (PeproTech, AF-200-21) to enhance peptide specific T-cell growth. Two days later, IL-2 (10 U per mL) was added. Every two days, half medium was replaced with fresh medium containing IL-2. After one week, the cultured T cells were stimulated again with DCs as described above. After a total of 3 weeks, CD8 and peptide-tetramer double-positive T cells were stained with PB conjugated anti-CD8 antibody (BD Biosciences Pharmingen, 558207) and PE-conjugated tetramer (Protein Chemistry Core-MHC Tetramer Lab in Baylor College of Medicine) and then sorted at MD Anderson Flow core facility. Sorted T cells were rested in medium overnight and expanded using a 14-day Rapid Expansion (REP) Protocol[4,62]. After expansion, the peptide-specific T cells were further characterised by flow cytometric analysis based on CD8 and tetramer staining. The gating and sorting strategies for FACS is shown in Supplementary Fig. 9.

**Caspase-3 cleavage based CTL killing assay**. T cell-mediated cell killing was analysed using a flow cytometry-based method by detecting T cell-induced caspase-3 cleavage in target cells[31]. The CCNI-ED10 peptide-reacting TIL2661, TIL2559, TIL2678 or Ted10 cells were thawed and cultured with 1000 U per mL of IL-2 overnight. $5 \times 10^6$ of target cells (T2, 293-A2 or melanoma cell lines) were labelled

with CellTrace^TM far red dye, DDAO-SE (Molecular Probes, C34553) at a final concentration of 0.6 μM for 15 min at 37 °C in 1 mL of PBS supplemented with 1% human serum. $5 \times 10^4$ DDAO-labelled target cells then were incubated in triplicates with different ratios of T cells for 3 h in 96 well plates. T cell-mediated caspase-3 cleavage was measured by intracellular staining with Cytofix/Cytoperm reagent (BD Biosciences, 554772) and PE conjugated anti-cleaved caspase-3 antibody (BD Bioscience, 550821) and the number of pre-apoptotic cells were determined by flow cytometry.

**cDNA constructs**. cDNAs for both wildtype and edited *CCNI* were cloned using the Gateway cloning system. Donor plasmids containing human non-edited *CCNI* cDNAs were purchased from Invitrogen. Site-directed mutagenesis (Clontech, 630703) was performed to produce edited cDNA and then cloned into a lentiviral vector, pHAGE (Addgene, 24526) by LR recombination (Thermo Fisher, 11791). All cDNA clones were verified by sequencing at the MD Anderson DNA core facility.

**Cell transfection**. In total $1 \times 10^6$ 293-A2 cells were seeded in each well of 6-well tissue culture plate in DMEM medium with 10% FBS to give 80% confluence on the day of transfection. For each well of cells, 2–4 μg of cDNA per 6 well and 8 μL of Lipofectamine 2000 (Invitrogen/Life Technologies, 11668-027) were used following manufacturers' instructions.

**ADAR1 knock down by small hairpin RNA (shRNA)**. pSIH-H1-GFP empty vector and pSIH-H1-GFP-ShADAR1 DNA were purchased from System Biosciences. To knock down *ADAR1* in melanoma cell lines, a Lentivirus was generated. $8 \times 10^6$ 293 cells were seeded in 100 mm plate until 80% of confluence. The 2nd generation lentiviral packaging plasmid pCMVR8.74 (Addgene, 22036) and PMD2G envelope expressing plasmid (Addgene, 12259) were co-transfected with pSIH-H1-GFP empty vector or pSIH-H1-GFP-ShADAR1 DNA as describe above. Supernatant containing virus was harvested at day 2 and day 3 after transfection. Melanoma cell lines were then transduced with filtered viral supernatant plus 10 μg per mL of polybrene (EMD Millipore Corp, TR-1003). Stably transduced cells were then selected based on expression of green fluorescent protein (GFP) and cell sorting after 4 days of transduction.

**Protein analysis by immunoblotting**. 293-A2 cells transfected with the indicated lentiviral expression vectors were lysed in RIPA cell lysis buffer (Thermo Fisher scientific, 89900), and cell lysates (1 μg per sample) were subjected to SDS–PAGE and transferred onto nitrocellulose blot membranes for immunoblotting using anti-CCNI antibody (1:2000 dilution, Sigma-Aldrich, GW22274). The same membrane was then striped and re-blotted with an antibody for the housekeeping protein actin as loading control (cell signalling, 4970, 1:2500 dilution).

**RT-PCR and PCR product sequencing**. Total RNA was isolated using Qiagen mini RNeasy kit (Qiagen 74104) and subjected to cDNA synthesis using a high-capacity cDNA kit (Thermo Fisher scientific, 4368813). Primers that flank the editing site of *CCNI* mRNA were used to amplify *CCNI* DNA fragment. *CCNI* PCR primers used were forward primer: 5′-CACTAGGGAAGCACAGATGTG-3′ and reverse primer: 5′- CCAATGGTGTGGCTGTGTGAAG-3′. PCR product was then purified using Qiagen QIAquick PCR Purification Kit (Qiagen, 28104). The purified PCR product was sequenced at the DNA sequencing core facility at MD Anderson.

**Quantitative PCR (qPCR)**. Total RNA was isolated and converted to cDNA as described above. qPCR was performed using iTaq™ Universal SYBR® Green Supermix reagent (Bio Red, 1725122) in a C1000^TM Thermal Cycler CFX96 Real-Time System following manufacturer's instructions (Bio-Rad). Primers used for amplification of *ADAR1* were 5′-GACACCGRCACTGCCACCTTC-3′ (forward) and 5′-GGTAGATACTCAGTTCCTGG-3′ (reverse). The house-keeping gene *GAPDH* was used for normalisation and amplified with 5′-CATCATCTCTGC CCCCTCT-3′ (forward) and 5′-GGTGCTAAGCAGTTGGTGGT-3′ (reverse).

**TA vector cloning**. PCR product that gave the *CCNI* R/G editing site was cloned into a TOPO TA cloning vector (Invitrogen,45-0030) following the manufacture's instruction. After transformation, plasmid DNA was isolated from 96 bacteria clones and sequenced at the Sequencing Core Facility of MDACC. The ratio of edited to non-edited *CCNI* clones was determined by DNA alignment.

**RNA sequencing of melanoma cell lines**. To analyse *CCNI* and other RNA editing events in melanoma cell lines, we performed RNA sequencing analysis. RNA was isolated from melanoma cell lines using RNeasy mini kit (Qiagen) and subjected to next generation RNA sequencing at the Deep Sequencing Core Facility of MDACC.

## Data availability

The TCGA data referenced during the study are available in a public repository from the Cancer Genomics hub (CGhub) website (http://cghub.ucsc.edu). The list of screened RNA editing sites is available as supplementary data. HLA ligandomics LC-MS/MS data supporting peptide sequence identifications has been deposited at PeptideAtlas with the dataset identifier PASS01150. RNA-seq data for determination of CCNI editing levels was made available at the Sequence Read Archive (SRA) under dataset identifier SRP154434.

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

## Acknowledgements

This work was supported by the following grants and foundations: Philanthropic contribution to University of Texas MD Anderson MoonShot-Talla for melanoma and EI Paso foundation (P.H.). Cancer Prevention Research Institute of Texas foundation (Immatics, DP150029). NIH R01 CA175486 (H.L.), U24CA209851 (H.L. and G.B.M.). CPRIT RP130397 (Y.C.). Philanthropic Bob Ladd funds (M.Z). Thanks to all members in Clinical TIL Production center of Department of Melanoma Medical Oncology MD Anderson Cancer Center for making TILs and tumour cell lines. Thanks to the bio-banking team at Immatics for sample acquisition, the Discovery lab team for preparation of tissues, DNA, RNA and peptides, the mass spec team for peptide analysis and validation, the CMC team for peptide synthesis and the bioinformatics team for data analysis. Thanks to Cassian Yee and the Lab members for providing the protocol and sharing their method of isolating and growing specific T cells from PBMCs.

## Author contributions

M.Z. and J.F. designed the study, performed experiments, data analysis, and wrote the manuscript; J.R., X.P., L.H. and H.L. performed RADAR and TCGA RNA-seq data analysis; A.T., Y.C. and G.L. performed initial peptide experiments; M.A.F., C.H. and C. B. generated TILs and cell lines; L.J.W. performed flow cytometry; O.S., H.S.-J. and T.W. managed the XPRESIDENT® program for mapping the human immunopeptidome covering sample acquisition, tissue preparation, peptide elution and mass spectrometry; C.-C.T. and V.G. performed the mass spectrometry data analysis; F.H. performed the proteogenomics data analysis; K.L.S., Y.-H.T. (S.T.), K.K. and X.X. performed DNA cloning; T.W., G.B.M. and P.H. supervised the work.

## Additional information

**Competing interests:** J.F., F.H., V.G., O.S., H.S.-J., C.-C.T. and T.W. are employees of Immatics. H.S.-J. and T.W. are shareholder of Immatics Biotechnologies. H.L. is a shareholder and scientific advisor of Precision Scientific and Eagle Nebula. P.H. is a Scientific Advisor for Immatics. The remaining authors declare no competing interests.



