## [Peer Review File · Nature Communications]

REVIEWERS' COMMENTS:

Reviewer #1 (Remarks to the Author):

The authors have addressed most of my technical comments. However, what remains is the generalizability of these findings.
Not sure they are really significant

Reviewer #2 (Remarks to the Author):

The manuscript by Minying Zhang et al, is really interesting and relevant.
In the revised manuscript, the authors have clearly replied to the comments and remarks. The final article is well written and highlights data that could be relevant in the field of oncology and immunooncology.

Reviewer #3 (Remarks to the Author):

The revised version of the manuscript "RNA editing-derived epitopes function as cancer antigens to elicit immune responses" by Zhang et al, addresses all my previous concerns. The work is novel, interesting, well written, has wide possible implications (including, potentially, clinical) and will be of interest to the wide community.

Reviewer #4 (Remarks to the Author):

The authors have gone a long way to answer the reviewers' comments, including conducting some experiments. They have, on the most part, sufficiently modified the manuscript to address my previous comments. I would like to recommend publication in Nature Communications but there are several unresolved issues that must first be addressed:

1) I appreciate the changes to the title and throughout the manuscript to clarify that the "edited peptides" identified are not tumor-specific. However, for this reason, calling these peptides "novel" (on page 1, line 16) may be misleading and should be omitted. Particularly because the search space was restricted to known sites present in the RADAR database. To further alleviate this, I suggest to indicate in the abstract that the identified edited peptides are "over-expressed self-antigens", as the authors note in their reply to the reviewers (or something similar along these lines).

2) Page 1 line 18-20: "... function as tumour antigens to elicit immune responses. We provide... these epitopes". This seems to be too generalized considering the support from only a single editing site. Please rephrase to state more explicitly that proof of concept is provided by a single example.

3) In continuation to my original major comment (bullet C), I still think a pivotal issue that has not been sufficiently explored and discussed in the manuscript is the possibility of these "over-expressed self-antigens" being highly expressed in some normal tissues as well. In support of this,

when combining the CCNI expression rates across normal tissues in GTEx and the A-to-I editing rates shown in the REDportal database for the CCNI-ED editing site, one finds solid grounds for concern that immunotherapies targeting CCNI-ED will be toxic to multiple tissues. In my original comment I mentioned brain tissues as potential candidates for toxicity, but this is just an example. For example, looking at artery tissues the GTEx and REDportal data reveals extremely high CCNI expression and 12-17% editing rate. Although the authors argue that one cannot compare GTEx to TCGA, the intra-GTEx data provides a clear enough picture that pressing safety issues exist.

Furthermore, the ULN is to some extent an arbitrary threshold. There is no evidence that CCNI-ED expression levels that are slightly to moderately under the ULN are insufficient to activate cytotoxic activity in the presence of TILs that target CCNI-ED.

If not addressed by the authors in a dedicated analysis, the safety concerns should be clearly mentioned in the Abstract and discussed in greater detail in the manuscript.

Minor:

Line 151: "result" -> "results"

We would like to thank all reviewers again for the great help and constructive comments of our manuscript. Please find below our response to Reviewer #4:

The authors have gone a long way to answer the reviewers' comments, including conducting some experiments. They have, on the most part, sufficiently modified the manuscript to address my previous comments. I would like to recommend publication in Nature Communications but there are several unresolved issues that must first be addressed:

1) I appreciate the changes to the title and throughout the manuscript to clarify that the “edited peptides” identified are not tumor-specific. However, for this reason, calling these peptides “novel” (on page 1, line 16) may be misleading and should be omitted. Particularly because the search space was restricted to known sites present in the RADAR database. To further alleviate this, I suggest to indicate in the abstract that the identified edited peptides are “over-expressed self-antigens”, as the authors note in their reply to the reviewers (or something similar along these lines).

Response: Following reviewers suggestion, we removed the word “novel” from abstract to avoid confusion. While the reported MHC peptides are “novel” in the sense that standard data processing of LC-MS data by protein database search would not identify these peptides, they are indeed not “novel” for the patient in the sense of a neoantigen. We also emphasized the aspect that RNA editing peptides are self-antigens.

2) Page 1 line 18-20: “... function as tumour antigens to elicit immune responses. We provide... these epitopes”. This seems to be too generalized considering the support from only a single editing site. Please rephrase to state more explicitly that proof of concept is provided by a single example.

Response: We agree that this might be misleading. The plural form was used since we characterized the two HLA-A*02-binding peptides CCNI-ED10 and -ED9 in-depth. We rephrased the abstract to avoid this confusion and emphasized that only CCNI edited peptides were deeply investigated.

3) In continuation to my original major comment (bullet C), I still think a pivotal issue that has not been sufficiently explored and discussed in the manuscript is the possibility of these “over-expressed self-antigens” being highly expressed in some normal tissues as well. In support of this, when combining the CCNI expression rates across normal tissues in GTEx and the A-to-I editing rates shown in the REDportal database for the CCNI-ED editing site, one finds solid

grounds for concern that immunotherapies targeting CCNI-ED will be toxic to multiple tissues. In my original comment I mentioned brain tissues as potential candidates for toxicity, but this is just an example. For example, looking at artery tissues the GTEx and REDportal data reveals extremely high CCNI expression and 12-17% editing rate. Although the authors argue that one cannot compare GTEx to TCGA, the intra-GTEx data provides a clear enough picture that pressing safety issues exist.

Furthermore, the ULN is to some extent an arbitrary threshold. There is no evidence that CCNI-ED expression levels that are slightly to moderately under the ULN are insufficient to activate cytotoxic activity in the presence of TILs that target CCNI-ED.

If not addressed by the authors in a dedicated analysis, the safety concerns should be clearly mentioned in the Abstract and discussed in greater detail in the manuscript.

Response: The ULN is supposed to assess the patient population that provides a potential therapeutic window. We showed the actual presentation levels of edited peptides on 421 normal samples of 35 different organs and compared that to levels in tumor tissue. The ULN is determined in a conservative fashion since it depends on the organ with the highest abundance rather than an average over all normals. Thus, on the therapeutically relevant measurement level (HLA peptide copies), the possibilities for healthy tissue peptide presentation were thoroughly investigated. In addition, there is no evidence of cytotoxic activity against healthy tissue since we show that T cells with reactivity against edited peptides are physiologically present in cancer tissue and thus in patients without evidence of severe side effects. But we agree that this is not evidence of safety of potential therapeutic approaches and that the analysis is strongly dependent on the set of healthy tissues included. We highlighted in the discussion that there is a need for further studies investigating safety using a more comprehensive set of tissues. We also updated the abstract to avoid the misconception that CCNI-ED can directly be used as immunotherapeutic target and elaborated on this opportunity and the safety concern in that regard in the discussion.

Minor:

Line 151: "result" -> "results"

We changed it.